# Ethanolic Extract of Duea Ching Fruit: Extraction, Characterization and Its Effect on the Properties and Storage Stability of Sardine Surimi Gel

**DOI:** 10.3390/foods12081635

**Published:** 2023-04-13

**Authors:** Natchaphol Buamard, Avtar Singh, Bin Zhang, Hui Hong, Prabjeet Singh, Soottawat Benjakul

**Affiliations:** 1International Center of Excellence in Seafood Science and Innovation (ICE-SSI), Faculty of Agro-Industry, Prince of Songkla University, Hat Yai 90110, Songkhla, Thailand; natchaphol.b@psu.ac.th (N.B.); avtar.s@psu.ac.th (A.S.); 2College of Food and Pharmacy, Zhejiang Ocean University, Zhoushan 316022, China; zhangbin@zjou.edu.cn; 3Beijing Laboratory for Food Quality and Safety, College of Food Science and Nutritional Engineering, China Agricultural University, Beijing 100083, China; honghuicau@126.com; 4College of Fisheries, Guru Angad Dev Veterinary and Animal Sciences University, Ludhiana 141004, Punjab, India; prabjeetsingh@gadvasu.in; 5Department of Food and Nutrition, Kyung Hee University, Seoul 02447, Republic of Korea

**Keywords:** Duea ching fruit, fig fruit, sardine surimi, gel properties, food packaging, shelf life

## Abstract

The quality of surimi gel can be improved using protein cross-linkers, especially from plant extracts. Apart from the presence of phenolic compounds, Duea ching fruit is rich in calcium, which can activate indigenous transglutaminase or form the salt bridge between protein chains. Its extract can serve as a potential additive for surimi. The effect of different media for the extraction of Duea ching was studied and the use of the extract in sardine surimi gel was also investigated. The Duea ching fruit extract (DCE) was prepared using distilled water and ethanol (EtOH) at varying concentrations. The DCE prepared using 60% EtOH (DCE-60) had the highest antioxidant activity and total phenolic content. When DCE-60 (0–0.125%; *w/w*) was added to the sardine surimi gel, the breaking force (BF), deformation (DF) and water holding capacity (WHC) of the gel upsurged and the highest values were attained with the 0.05% DCE-60 addition (*p* < 0.05). However, the whiteness of the gel decreased when DCE-60 levels were augmented. The gel containing 0.05% DCE-60, namely D60-0.05, showed a denser network and had a higher overall likeness score than the control. When the D60-0.05 gel was packed in air, under vacuum or modified atmospheric packaging and stored at 4 °C, BF, DF, WHC and whiteness gradually decreased throughout 12 days of storage. However, the D60-0.05 gel sample showed lower deterioration than the control, regardless of the packaging. Moreover, the gel packaged under vacuum conditions showed the lowest reduction in properties throughout the storage than those packaged with another two conditions. Thus, the incorporation of 0.05% DCE-60 could improve the properties of sardine surimi gel and the deterioration of the resulting gel was retarded when stored at 4 °C under vacuum packaging conditions.

## 1. Introduction

Fig trees are abundant in southwest Asia, southeast Asia and the Mediterranean, and their fruits are consumed as whole fruit or processed as juice, jam, etc. Duea ching (*Ficus botryocarpa* Miq.) is one of the fig cultivars found in southern Thailand [1]. It is rich in vitamins, minerals, dietary fiber, as well as amino acids. In addition, the fruit is low in sodium, but high in calcium, potassium, magnesium, iron and zinc [2]. Additionally, numerous bioactive compounds, e.g., polyphenols and carotenoids, etc. are also present in figs. Those compounds are known to have several health benefits, including anti-inflammatory, antioxidant, as well as anti-bacterial potential [3,4,5]. Moreover, the polyphenols and pigments can act as protein cross-linkers, which can strengthen the protein network, particularly in muscle proteins [4]. The cross-linking ability of plant extracts, such as coconut husk, cluster bean and pineapple peel, toward fish myofibril proteins, is related with the increased gel strength of fish mince or surimi from different species [6,7,8].

The gelation of myofibrillar proteins is compulsory for the formation of acceptable products from washed fish mince, or so-called surimi [5,9]. Dark-fleshed surimi gels, such as those from sardine and mackerel, undergo proteolytic activity due to indigenous proteases, which are involved in gel weakening caused by high levels of proteases [6]. Some indigenous proteases in fish muscles can be activated at high temperatures (55–65 °C) [10]. On the other hand, endogenous transglutaminase (TGase) is a calcium (Ca^2+^) activated enzyme that can be found in fish muscles. TGase has been known to cross-linkmuscle proteins via the formation of ε-(γ-glutamyl) lysine linkage. It enhances the setting phenomenon, leading to the improved gel strength [6,9]. To strengthen surimi gel, various cross-linkers, especially microbial TGase, plant polyphenols, oxidized phenolics, etc., have been employed [6,11,12,13].

The food industry has established several packaging technologies aimed at prolonging the shelf life of perishable products, especially fish and fish products. Vacuum packaging (VAC) and modified atmosphere packaging (MAP) have been widely used to maintain the quality of various food products by retarding microbial growth and preventing post-contamination, thus extending the shelf life [14]. In general, MAP extends the shelf life, but the temperature and initial microbiological load of the raw material determine its efficiency [14]. VAC is performed by removing the air or oxygen from the package [15]. Oxygen removal can retard the growth of aerobic spoilage bacteria, thereby prolonging the shelf life of products [15]. Although VAC and MAP have been widely used in food packaging, very few reports are available on their application in preserving surimi gel.

Protein cross-linkers, mainly from plant extracts, particularly those that are rich in calcium, such as Duea ching fruit, could be an alternative additive to induce protein polymerization of surimi from dark-fleshed fish possessing poor gel-forming properties. No information on the uses of the Duea ching extract (DCE) in surimi gel exists. Its impact on storage stability has not been investigated. This study, therefore, aimed to investigate the characteristics of DCE, the surimi gel-strengthening effect and its impact on the quality of the gel from sardine surimi under different packaging conditions, during extended refrigerated storage.

## 2. Materials and Methods

### 2.1. Chemicals and Materials

All the chemicals were purchased from Sigma-Aldrich, Inc. (St. Louis, MO, USA) and Loba Chemie Pvt. Ltd. (Mumbai, Maharashtra, India). Frozen sardine surimi (grade A) supplied by Chaicharoen Marine (2002) Co., Ltd. (Pattani, Thailand) was used.

### 2.2. Effect of Extraction Media on Compositions and Antioxidant Activities of Fig (Duea Ching) Extract

#### 2.2.1. Preparation of Fig Extract Using EtOH at Different Concentrations

Fig fruits (Figure 1) were obtained from a local garden in Trang province, which is located in the southern part of Thailand. The collected fruits were immature, having greenish peel and a firm texture, without apparent damage. The fruits were cleaned with tap water. The samples with 86.9% moisture content, as measured by the AOAC method [16], were blended to homogeneity. Then, the fig paste was subjected to extraction with water or EtOH at varying concentrations (20–100%, *v/v*) using a paste/EtOH ratio of 1:15 (*w/v*) [17]. After stirring using a magnetic stirrer (IKA-Werke, Staufen, Germany) for 1 h in the dark, the mixtures were subsequently centrifuged for 30 min (5000× *g*; 4 °C). Filtration was adopted for the supernatants using Whatman filter paper No. 1. The EtOH in the obtained filtrates was removed at 40 °C using an Eyela rotary evaporator (Tokyo Rikakikai, Co., Ltd., Tokyo, Japan), followed by nitrogen purging. The remaining supernatants were collected and frozen at −20 °C before lyophilization with the aid of a Scanvac Model Coolsafe 55 freeze dryer (Coolsafe, Lynge, Denmark). The obtained extracts were placed in an amble bottle, capped tightly and stored in a desiccator under dark condition. The dried extracts prepared using distilled water and EtOH at concentrations of 20, 40, 60, 80 and 100% were referred to as DCE-W and DCE-20, DCE-40, DCE-60, DCE-80 and DCE-100, respectively. All the extracts were analyzed.

#### 2.2.2. Analyses

##### Total Phenolic Content (TPC)

The TPC of the DCEs was determined using the Folin–Ciocalteu method, as tailored by Mittal et al. [18]. The TPC was reported as mg gallic acid equivalents (GAE)/g dry extract.

##### Calcium Content

An inductively coupled plasma optical emission spectrometer (ICP-OES) (Avio 500, PerkinElmer, Shelton, CT, USA) was used for calcium determination, as described by Wijayanti et al. [19]. Calcium was detected at a wavelength of 317.933 nm.

##### Antioxidant Activities (AOX)

The methods given by Singh et al. [20] were employed for testing the AOX, including DPPH radical-scavenging activity (D-RSA), ABTS radical-scavenging activity (A-RSA), ferric reducing antioxidant power (FRAP) and metal chelating activity (MCA).

For the D-RSA, the sample (1.5 mL) was combined with 1.5 mL of 0.15 mM DPPH in 60% ethanol. The mixture was mixed and allowed to stand at room temperature in the dark for 30 min. The absorbance of the resulting solution was measured at 517 nm, using a spectrophotometer (UV-1800, Shimadzu, Kyoto, Japan). A sample blank was prepared in the same manner, except that the corresponding solvents were used instead of the DPPH solution. A standard curve was prepared using Trolox in the range of 10–60 μM. The activity was calculated after subtraction of the sample blank.

For the A-RSA, the stock solutions included 7.4 mM ABTS solution and 2.6 mM potassium persulfate solution. The working solution was prepared by mixing the two stock solutions in equal quantities and allowing them to react for 12 h at room temperature in the dark. The solution was then diluted by mixing 1 mL of ABTS solution with 50 mL of methanol, in order to obtain an absorbance of 1.1 ± 0.02 at 734 nm using the spectrophotometer. The sample (150 μL) was mixed with 2850 μL of the ABTS solution, and the mixture was left at room temperature for 2 h in the dark. The absorbance was then measured at 734 nm using a spectrophotometer. A sample blank was prepared in the same manner, except that methanol was used instead of the ABTS solution.

For the FRAP assay, stock solutions including 300 mM acetate buffer (pH 3.6), 10 mM TPTZ (2,4,6-tripyridyl-s-triazine) solution in 40 mM HCl, and 20 mM FeCl_3_·6H_2_O solution were prepared. A working solution was freshly prepared by mixing 25 mL of acetate buffer, 2.5 mL of TPTZ solution and 2.5 mL of FeCl_3_·6H_2_O solution. The mixed solution was incubated at 37 °C for 30 min in a water bath (Memmert, D-91126, Schwabach, Germany) and was referred to as the FRAP solution. A sample (150 μL) was mixed with 2850 μL of the FRAP solution and kept for 30 min in the dark at room temperature. The ferrous tripyridyltriazine complex (colored product) was measured by reading the absorbance at 593 nm. A sample blank was prepared by substituting the distilled water for FeCl_3_ in the FRAP solution. The standard curve was prepared using Trolox ranging from 50 to 600 μM.

To determine the MCA of the DCE, the sample (940 μL) was mixed with 20 μL of 2 mM FeCl_2_ and 40 μL of 5 mM ferrozine. The reaction mixture was allowed to stand for 20 min at room temperature. The absorbance was then read at 562 nm. The blank was prepared in the same manner, except that distilled water was used instead of the sample. For the sample blank, the distilled water was substituted for the FeCl_2_ solution. The standard curve was prepared using the EDTA ranging from 10 to 60 μM.

The D-RSA, A-RSA and FRAP were expressed as µmol Trolox equivalents (TE)/g sample, while the MCA was reported as µmol EDTA equivalents (EE)/g sample.

##### LC/MS Profiling and Identification

The DCE showing the highest TPC and AOX (DCE-60) was subjected to identification using liquid chromatography–mass spectrometry (LC/MS). The qualitative identification of the compounds in DCE-60 was conducted using LC–quadrupole time-of-flight MS (LC-QTOF MS; 1290 Infinity II LC-6545 Q-TOF, Agilent Technologies, Santa Clara, CA, USA), as per the method given by Tagrida and Benjakul [17].

### 2.3. Effect of DCE on the Properties of Sardine Surimi Gel

#### 2.3.1. Preparation of Surimi Gel with DCE at Different Levels

After defrosting the surimi in a cold room overnight to obtain a core room temperature of 0–2 °C, the surimi was chopped in the presence of 2.5% (*w/w*) salt for the solubilization of the myofibrillar proteins. Chopping was conducted using a mixer (National Model MK-5080M, Selangor, Malaysia) for approximately 2 min. Thereafter, the DCE-60 was mixed with surimi paste at varying levels (0.025, 0.050, 0.075, 0.10 and 0.125%; *w/w*). The moisture content of the surimi paste was adjusted to 80% with water, followed by chopping for another 3 min. The surimi paste was subsequently stuffed into polyvinylidene chloride casing (2.5 cm diameter). All the operations were conducted at a temperature below 10 °C. The setting was carried out (40 °C, 30 min), followed by cooking (90 °C, 20 min). Finally, all the obtained gels were cooled in iced water for 30 min. The gels were left at 4 °C for 20 h before the determinations.

#### 2.3.2. Analysis

##### Breaking Force (BF) and Deformation (DF)

The BF and DF of the surimi gels were examined using a texture analyzer (Model TA-XT2, Stable MicroSystems, Surrey, UK) equipped with a spherical plunger (diameter 5 mm), which was pressed into the cut surface of the gel sample perpendicularly at a constant depression speed (60 mm/min) [21].

##### Expressible Moisture Content (EMC) and Whiteness

The gel samples were measured for EMC and whiteness, as per the methods of Buamard and Benjakul [6].

##### Textural Profile Analysis (TPA)

The TPA of the surimi gel samples was conducted using a texture analyzer (Model TA-XT2, Stable MicroSystems, Surrey, UK) with a P/50 cylinder probe. The operation was performed at a test speed of 5 mm/s. The hardness, springiness, cohesiveness, gumminess and chewiness were recorded [22].

##### Sensory Evaluation

The gel samples were subjected to sensory analysis using eighty non-trained panelists. The color, taste, texture and overall liking of the gel samples were evaluated using a 9-point hedonic scale [23].

##### Microstructure

The gel samples, including gel without and with the addition of DCE-60 at 0.025, 0.05 and 0.125 were prepared. All the samples were dehydrated using 25, 50, 70, 80, 90 and 100% serial EtOH dilution. The samples were critical point dried, using CO_2_ as the transition fluid. The prepared samples were mounted on a bronze stub and sputter coated with gold. The microstructure was visualized using a scanning electron microscope (SEM, Quanta 400, FEI, Eindhoven, The Netherlands) at a magnification of 10,000×.

### 2.4. Effect of DCE on Microbial Load, Lipid Oxidation and Properties of Surimi Gel Stored under Different Packaging Conditions during Refrigerated Storage

#### 2.4.1. Preparation of Gels Containing DCE Stored under Different Packaging Conditions

The surimi gels without and with DCE-60 at the level yielding the highest DF (0.05%) were prepared. The gels were then packaged separately in: (1) a polypropylene zip lock bag (atmosphere condition; ATM), (2) a nylon/LLDPE bag under vacuum packaging conditions (VP) and (3) a nylon/LLDPE bag under modified atmosphere packaging conditions (60% CO_2_: 10% O_2_: 30% N_2_; MAP). The gel samples were taken every 2 days for a total of 12 days and analyzed.

##### Gelling Properties

The stored samples were determined for the BF, DF, EMC and whiteness, as detailed above.

##### Lipid Oxidation and Microbiological Analyses

The peroxide value (PV) and thiobarbituric acid-reactive substances (TBARS) were determined [24] and used as the indices for lipid oxidation. For microbiological quality, the total viable count (TVC) and psychrophilic bacteria count (PBC) were quantified using the standard spread plate method [21].

### 2.5. Statistical Analysis

A completely randomized design (CRD) was used throughout the study. All experiments and analysis were carried out in triplicate. The data were subjected to analysis of variance (ANOVA). A comparison of means was conducted using the Duncan’s multiple range test [24]. SPSS for Windows (SPSS Inc., Chicago, IL, USA) was used for the statistical analysis. Data with a *p* < 0.05 were considered to be statistically significant.

## 3. Results and Discussion

### 3.1. Composition and Antioxidant Activities of Different Duea Ching Extracts

#### 3.1.1. Total Phenolic Content (TPC)

The TPC of the DCEs prepared using distilled water and EtOH at several different concentrations is shown in Table 1. The TPC was increased by augmenting EtOH concentrations up to 60%. Thereafter, the TPC was decreased when the EtOH was above 60% (*p* < 0.05). The highest TPC was found in DCE-60 (722.53 mg GAE/g dry extract), whereas the lowest content was obtained for DCE-W (310.19 mg GAE/g dry extract) and DCE-100 (308.51 mg GAE/g dry extract) (*p* < 0.05). Nevertheless, a similar TPC between DCE-W and DCE-100 was observed (*p* > 0.05). The polarity of solvents mainly affects the extraction efficacy of phenolics from the plant origin [17]. Buamard and Benjakul [6] also noticed the increasing TPC content of coconut husk extracts, when the EtOH concertation was increased to 60%. In addition, the source, age, processing conditions, temperature and type and concentration of the solvents also affect the extraction of phenolic compounds [6,22,23,24]. For example, the appropriate EtOH concentration for garlic husks was 50%, whereas a wider range of concentrations (40–90%) was suitable for the extraction of black currants [24,25]. Similarly, when garlic peel extracts were prepared using 80% methanol, a higher TPC content was obtained as compared to the 60% EtOH as the solvent [26]. Hence, the 60% EtOH yielded the DCE with the maximum amount of phenolic compounds.

#### 3.1.2. Calcium Content

The calcium content of the DCE extracts was slightly affected by the EtOH concentrations, as shown in Table 1. The calcium content was in the range of 1082–1099 mg/kg dry extract. Among all the extracts, DCE-60 had the highest calcium content (1099.17 mg/kg dry extract), followed by DCE-40 and DCE-80 (1092.12 mg/kg dry extract) (*p* < 0.05). The lowest content was attained in the extracts using distilled water (DEC-W) (*p* < 0.05). Nonetheless, the remaining samples had similar values (*p* > 0.05). Fig is a promising source of trace minerals, such as calcium, which provides strength to the fruits and allows it to maintain its weight [27,28,29]. Ahuja [30] documented that the calcium content in figs is 350 and 1620 mg per kg fresh and dry fruit, respectively, which is higher than that of the DCE. This could be due to the varying extraction techniques used. Moreover, the initial content of calcium might be different, due to the varying cultivars, climates, etc. Hence, the EtOH at 60% yielded the extract with the highest calcium content, which could be used to strengthen surimi or mince gels. Ca^2+^ has been reported to induce indigenous TGase [10,11]. In addition, it can strengthen the surimi gel via the ‘salt bridge’ [31]. Thus, DCE could serve as a source of calcium, apart from the phenolic compounds, used for protein cross-linking.

#### 3.1.3. Antioxidant Activities (AOX)

The AOX of the extracts from Duea ching fruits are shown in Table 1. The A-RSA, D-RSA, FRAP and MCA were increased by augmenting the concentrations of EtOH. The highest values determined by all the assays used were found when 60% EtOH was used as the extracting media (*p* < 0.05). Nevertheless, with a further upsurge in the EtOH concentration, all the AOX were decreased (*p* < 0.05). Among all the extracts, the DCE-W and DCE-100 extracts exhibited the lowest activities (*p* < 0.05). The A-RSA and D-RSA determine the potential of the compound to stabilize free radicals via donating a hydrogen atom in the amphiphilic and lipophilic system, respectively [32]. The FRAP measured the power of the antioxidants to reduce the Fe(III) to Fe(II)-TPTZ complex, whereas the MCA indicated the capability of the compounds to chelate the prooxidative metal ions [32,33]. The increasing AOX with the augmenting EtOH levels was more likely related with the higher release of phenolic compounds, including anthocyanins and flavonoids from the Duea ching fruit [34]. The result coincided with the TPC of the extracts. Similar results were documented by Tagrida et al. [17] and Buamard and Benjakul [6], when EtOH was used to prepare the extracts from betel/chaphlu leaves and coconut husk, respectively. The phenolic compounds possessed excellent AOX, due the presence of a large number of hydroxyl groups acting as radical scavengers [35,36]. Moreover, polyphenols consist of several aromatic rings and hydroxy groups, serving as the chelator towards metal ions. Furthermore, they can reduce the highly reactive oxidized form of metal ions to the lower reactive counterpart [33]. Additionally, the tocopherols, ascorbates, several pigments, carotenoids, etc., present in fig fruits could contribute to the AOX [34].

#### 3.1.4. Identification and Profiling of Compounds in Selected Duea Ching Extracts (DCE-60)

The compounds present in DCE-60 determined using the negative ion and positive ion mode of the LC/MS are shown in Table 2. In the negative ion mode, naringenin-7-O-glucoside (flavanone) was the most abundant compound. Quercetin 3-galactoside (hyperoside; quercetin), rutin (flavonoid glycoside) and indole-4-carbaldehyde (indole) were also present. In the positive ion mode, around five compounds were detected. However, rutin and quercetin were also noticed in the latter ion mode. In general, the positive ion mode charged analyte via protonation, whereas the deprotonation of analyte happened in the negative ion mode [37]. The former mode has been used widely, due to its high sensitivity as compared to the latter mode [37]. Naringenin-7-O-glucoside, also called prunin, is widely extracted from citrus fruits [38,39]. Quercetin 3-galactoside or hyperoside is a quercetin with a beta-D-galactosyl residue attached at position 3. It belongs to a group of plant pigments called flavonoids that give color to fruits, flowers and vegetables. Similarly, rutin is a plant pigment/flavanol glycoside between quercetin and disaccharide rutinose (α-l-rhamnopyranosyl-(1→6)-β-d-glucopyranose) [40]. All these compounds possess antioxidant, anticancer and anti-inflammatory properties, improve blood circulation and insulin resistance, promote colonic healing, etc. [23,40,41,42]. Indole-4-carbaldehyde is a hetero-arenecarbaldehyde indole, in which the hydrogen at position 4 is replaced by a formyl group. It has been used as antibacterial and antifungal agent [43]. Along with several bioactivities, these phenolic compounds can be used to improve the interactions among the food proteins by acting as a cross-linker. For example, rutin was incorporated into soy protein films as a protein cross-linker, which improved the film forming properties [44]. In addition to these compounds, chlorogenic acid, quinic acid, 4-acetoxyphenol and naringenin were also present and they provided several activities in the DCE-60 extract. Overall, DCE-60 contained several bioactive compounds, which could function as a protein cross-linker, an antioxidant and could be used for the preservation of food or food products.

### 3.2. Effect of DCE-60 at Various Levels on the Textural and Sensory Properties of Sardine Surimi Gel

#### 3.2.1. Breaking Force (BF) and Deformation (DF)

The BF and DF of the sardine surimi gel incorporated with DCE-60 at various levels are depicted in Figure 2A,B, respectively. The gel samples containing DCE-60 at all concentrations had the higher BF than the control gel (without DCE-60) (*p* < 0.05). The gel containing 0.05% DCE-60 had the highest BF (*p* < 0.05), in which the BF was increased by 100% compared to that of the control. However, when the concentrations were increased further, a lower BF was noticed (*p* < 0.05). The same trend was noticed for the DF, where the highest increase (by 50%) was attained in the presence of 0.05% DCE-60. Generally, the BF and DF represent the strength and elasticity of the gel, respectively, which are determined by the formation of an ordered network by myofibrillar proteins via various bondings [6]. Myofibrillar proteins consist of myosin heavy chain (MHC) and actin as the major components. MHC is the main protein responsible for gelation [10,45]. The upsurged BF/DF might be attributed to the greater cross-linking of myofibrillar proteins caused by phenolics in DCE-60. Hydrogen bonding between the hydroxyl group of polyphenols and the hydrogen acceptor localized in the protein chains could also enhance the protein–phenolic compound interactions, thus causing higher gel strength [3,9]. Moreover, the presence of calcium could also act as the salt bridge between the protein chains, as well as enhance the activity of TGase [31]. Nevertheless, when the concentration of DCE-60 was added at levels higher than 0.05%, the excessive cross-linking in forms of aggregation and coagulation of the myofibrillar proteins might lead to the development of an unordered gel network. Similar results were noticed when oxidized phenolic compounds or coconut husk extracts were incorporated into surimi gel [6,9,13]. Additionally, the self-aggregation of phenolic compounds in DCE-60 at higher concentrations could be another possible reason for lowering the protein cross-linking ability in gel than those with DCE-60 at a lower concentration. Plant extracts have been used for surimi gel strengthening due to the presence of phenolic compounds. Ethanolic extracts from pomegranate and pineapple at an optimum level could improve the gelling property of surimi from silver carp [8,46]. Hence, DCE-60 at 0.05% could be used as a gel enhancer in sardine surimi.

#### 3.2.2. Expressible Moisture Content (EMC)

The EMC of the gel without and with DCE-60 at varying concentrations are shown in Figure 2D. With the addition of DCE-60, the EMC was decreased with the upsurge in DCE-60 concentration up to 0.05%, which yielded the lowest MC value (*p* < 0.05). Nevertheless, with a further upsurge in the DCE-60 levels, the EMC was increased (*p* < 0.05). A similar EMC was achieved between the control and gel with 0.125% DCE-60 added to the gel samples (*p* > 0.05). Moreover, a similar EMC was observed between the samples with 0.025 and 0.1% DCE-60 added (*p* > 0.05). The EMC represents the WHC of the gel, which are inversely proportional to each other. The lower the EMC of the gel with 0.05% DCE-60 added, the higher the WHC was achieved. This might be attributed to the formation of the ordered three-dimensional gel network, which was able to imbibe more water [6,45]. Normally, gel with higher BF has the ability to hold more water. The data was also in tandem with the maximum BF of the gel containing 0.5% DCE-60 (Figure 2A). A similar result was reported by Buamard and Benjakul [6] when coconut husk at an appropriate level was incorporated into sardine surimi gel.

#### 3.2.3. Whiteness

The gel whiteness was decreased continuously as the DCE-60 levels increased (Figure 2D). The highest whiteness value was found in the control, whereas the samples with 0.1 or 0.125% DCE-60 added had the minimum value (*p* < 0.05). This decrease was plausibly due to the green color of DCE-60, which was increased with increasing EtOH concentration. In general, the type and source of the additives and the presence of the pigment in the additive have an influence on the color of the resulting surimi gel [45]. The color of surimi gel could be improved by several whitening agents, oil, etc., which could enhance the light scattering properties of the gel [45,47]. The color of the extract reduced the whiteness of the surimi gel. Therefore, dechlorophilization of the fig extract should be performed [23].

#### 3.2.4. Textural Properties

The textural properties, as determined by the TPA of all the gel samples without and with DCE-50 at different amounts, are varied (Table 3). Hardness, the force required to compress the sample to attain a given deformation of the sample, was in agreement with the BF (Figure 2A). The hardness was increased when the DCE-60 amount was increased and the maximum value was obtained at 0.05% (*p* < 0.05). However, no differences in the hardness between the control and samples with 0.1% DCE-60 added were found (*p* > 0.05). For springiness, the highest value was found in the sample containing 0.05% DCE-60 (*p* < 0.05). A similar springiness was attained between the control and the samples incorporated with 0.125% DCE-60. In addition, similar values were found between samples containing 0.025 and 0.1% DCE-60 (*p* > 0.05). The springiness elucidates the rubberiness of the gel and its capability to spring back after first bite deformation [48]. A similar trend was noticed for cohesiveness, which measured the force required to overcome the internal bonding of the material [48]. The gumminess and chewiness almost had a similar trend to the hardness, where the highest values were noticed for the gel with 0.05% DCE-60 added (*p* < 0.05). Chewiness is the energy required to chew the gel to the point where it can be swallowed, and gumminess represents the energy used for the swallowing of the semisolid food [49]. Consequently, DCE-60 at the appropriate level, such as 0.05%, could be added without compromising the textural properties.

#### 3.2.5. Acceptability

No differences in the acceptability scores for appearance, color and taste, among the control and samples with 0.025, 0.05 and 0.075% DCE-60 added were detected (*p* < 0.05) (Table 4). All the above-mentioned samples had a higher acceptability score for appearance, color and taste than the gels with 0.10 and 0.125% DCE-60 added (*p* < 0.05). This could be owing to the augmenting color and stringent taste of the extract caused by the presence of higher amounts of polyphenols when the concentration of DCE-60 was increased. However, all the samples had a similar odor score (*p* > 0.05). Nevertheless, the samples incorporated with 0.05 and 0.075% DCE-60 had the highest score for texture (*p* < 0.05). A similar likeness score for texture was obtained between the control and gel with 0.025, 0.10 and 0.125% DCE-60 added (*p* > 0.05). The result coincided with hardness and BF scores (Table 3 and Figure 2A). Therefore, the maximum level, which could be added to sardine surimi without affecting the acceptability of the sardine surimi gel was 0.05%. Such a level did not alter the color, taste and odor of the sardine surimi gel. The result was substantiated by the similar overall likeness score for the gel containing 0.05% DCE-60, which was similar to the gel with 0.075% DCE-60 added.

#### 3.2.6. Microstructure

The microstructures of the surimi gel without and with 0.025, 0.05 and 0.125% DCE added, determined using SEM, are shown in Figure 3A–D. The control gel had an irregular and looser network with larger voids as compared to the other samples (Figure 3A). However, with the addition of DCE-60, the gel network became denser with higher connectivity, especially for the gel incorporated with 0.05% DCE-60. The gel containing 0.05% DCE-60 had smaller voids and thicker proteins strands (Figure 3C). This could be due to the cross-linking ability of the phenolics and calcium present in DCE-60, which enhanced the interactions or aggregation among the myofibrillar proteins. When 0.125% DCE-60 was added, the size of the void increased and protein agglomeration was noticed (Figure 3D). This could be associated with excessive and fast protein cross-linking, which led to the formation of a coagulated protein network. The result was in line with the BF of the gel sample (Figure 2A). Hence, the DCE-60 concentrations had an impact on the gel microstructure and DCE-60 as an appropriate concentration could enhance the interactions among the myofibrillar proteins, building up the ordered gel network.

The DCE-60 at 0.05% (D60-0.05), rendering the higher gel properties and consumer acceptability, was selected for gel preparation. The resulting gels were stored under varying packaging conditions at 4 °C in comparison to the control (without DCE-60).

### 3.3. Effect of DCE-60 Addition on the Microbial Load, Lipid Oxidation and Gel Properties of Sardine Surimi Gel Packed under Varying Conditions during Refrigerated Storage

#### 3.3.1. TVC and PBC

The TVC and PBC of the gel samples without and with 0.05% DCE-60 (D60-0.05) packed under varying conditions during storage are shown in Table 5. Regardless of the packaging conditions, the TVC and PBC were not detected in both samples (control and D60-0.05) at day zero, which suggested the hygienic nature of the processing conditions. In addition, the heating of the surimi paste in casing occurred mimicking the sous vide process, which has been reported to be effective in inactivation of microorganisms [15]. When the storage time was increased, the TVC and PBC upsurged for both samples. However, the control showed the higher increase for both the TVC and PBC in ATM packaging (*p* < 0.05). DCE-60 was able to prolong the shelf life of the sardine surimi gel to day 8 as compared to control, in which the acceptable limit for both the TVC and PBC (5 log CFU/g) was exceeded at day 6 (*p* < 0.05) [50]. The phenolic extract from *Perilla frutescens* leaf was found to inhibit bacterial growth and extended the shelf life of *Argyrosomus argentatus* surimi fish balls at 4 °C up to 12 days [51].

Similar results were noticed in VAC packaging conditions, except the shelf life was extended to day 10 for the control, but the sample with 0.05% DCE-60 added had a TVC lower than the limit of up to 12 days. For D60-0.05, no TVC and PBC were observed for day 0 and 2, as compared to the control gel. Overall, the higher TVC and PBC values were obtained for the control gel as compared to the D60-0.05 sample (*p* < 0.05). At day 12, the control had 5.17 and 4.98 log CFU/g of TVC and PBC, respectively. For the D60-0.05 sample, the TVC and PBC were found to be 4.59 and 4.37 log CFU/g, respectively, on day 12, which was still under the acceptable limit.

For MAP, a similar trend was obtained to the VAC packaging condition, except for the control sample, whose TVC and PBC limit was exceeded after day 8 and 10 for the control and D60-0.05 sample, respectively. The TVC/PBC for the control and D60-0.05 sample at day 10 and 12 were 5.86/5.39 and 5.80/5.74 log CFU/g sample, respectively. The lower microbial count in the D60-0.05 gel samples was more likely associated with the antimicrobial activity of several polyphenols and other bioactive compounds present in the DCE-60 extract. Plant extracts have been known to possess excellent antimicrobial activity. An EtOH extract from betel leaves (400 and 600 ppm) extended the shelf life of Nile tilapia for 12 days at 4 °C by inhibiting several spoilage bacteria [22].

Irrespective of the samples, the VAC conditions resulted in the lowest TVC and PBC, followed by MAP during storage (*p* < 0.05). The sample kept in the ATM had the maximum microbial load, as witnessed by the higher TVC/PBC values during storage. The removal of oxygen from the package under vacuum conditions could retard the proliferation of aerobic spoilage bacteria. For MAP packaging, O_2_ was reduced and at a lower level than that found in the atmosphere [15]. When VAC and MAP conditions were compared, the higher microbial load in the latter was more likely associated with presence of O_2_ (10%). Although the presence of CO_2_ can retard microbial growth [52], microbial growth could inevitably take place as compared to VAC packaging where oxygen/air has been removed almost completely. Hence, the incorporation of 0.05% DCE-60 in surimi gel in combination with VAC packaging could retard microbial growth during storage.

#### 3.3.2. PV and TBARS

The PV and TBARS representing the primary and secondary oxidation products in the oxidized lipid are shown in Table 5. Both values were augmented with an upsurge in the storage time; however, the higher increase was noticed for control gels as compared to the D60-0.05 sample, regardless of the packaging conditions (*p* < 0.05). This could be due to the AOX of the DCE-60 (Table 1), which prevented lipid oxidation during the processing and storage of surimi gel. Lipid oxidation generates products, which can cause the off-flavor and off-odor of seafoods. This is mainly caused by the oxidation of polyunsaturated fatty acids, thereby shortening their shelf life [53]. Among the packaging conditions, ATM had the highest PV and TBARS, followed by MAP (*p* < 0.05). Overall, the lowest lipid oxidation was noticed in the VAC samples. This could be due to the absence of oxygen, which could be involved in the lipid oxidation process by the formation of peroxyl radicals [54]. Therefore, the addition of 0.05% DCE-60 to sardine surimi gel packed under different conditions was able to lower the lipid oxidation in the surimi gel during storage at 4 °C.

#### 3.3.3. BF and DF

The BF and DF of both samples (control and D60-0.05) were decreased with augmenting storage time, irrespective of the packaging conditions (*p* < 0.05) (Table 6). On day 8, when the D60-0.05 sample had a TVC under the limit, the BF and DF was decreased by 36 and 31%, respectively. Based on the microbial count, the gel samples packed under VAC showed extended shelf life for 12 days, where the BF and DF of the control/D-60-0.05 samples were decreased by 30/21% and 24/7.8%, respectively. Similarly, for the MAP samples, a 29 and 15% reduction in the BF and DF was noticed on day 10, respectively. When compared to the samples, the D60-0.05 gels had a lower decrease in the gel properties with a higher shelf life than the control gel sample (*p* < 0.05). A decreased BF and DF were plausibly attributed to a degradation of the myofibrillar caused by psychrophilic bacteria that proliferated during storage [55]. In addition, the radicals generated during lipid oxidation may oxidize the myofibrillar proteins, causing the fragmentation of the proteins in gel. Singh et al. [21] reported a reduction in the gel properties of sardine surimi gel kept at 4 °C. When comparing different packaging conditions, VAC had the least reduction in BF/DF at the end of the storage. Moreover, the D60-0.05 samples were preserved for 12 days. This could be because of the lower microbial count and lipid oxidation in VAC samples, as compared to the samples kept under other packaging conditions (Table 5).

#### 3.3.4. EMC

Changes in the EMC of the control and D60-0.05 samples packed under varying conditions during storage at 4 °C are presented in Table 6. Both samples showed increasing EMC under all packaging conditions as the storage time increased. However, the D60-0.05 samples packed under VAC had the lowest changes in the EMC during storage (*p* < 0.05), whereas the highest increase was noticed in samples packed under ATM (*p* < 0.05). Water loss during storage was commonly found when the gel structure was hydrolyzed or deformed. The highest destruction in the three-dimensional gel network was noticed in the ATM and MAP packed samples, especially in the control as evidenced by the decreasing BF and DF (Table 6). Such a gel could not imbibe water effectively. This could be due to the higher PBC and lipid oxidation in the control (Table 5). Hence, 0.05% DCE in combination with VAC packaging lessened the water loss in the gel samples during the extended storage at 4 °C.

#### 3.3.5. Whiteness

Regardless of the samples, the whiteness of samples was slightly decreased as the storage time was augmented (*p* < 0.05) (Table 6). Overall, the control had the highest whiteness than the D60-0.05 sample throughout storage (*p* < 0.05), mainly mediated by the color and indigenous pigments present in extract. For the ATM and MAP samples, the whiteness was decreased after day 2 of storage. On the other hand, the VAC sample showed a reduction in whiteness after day 4 (*p* < 0.05). Thereafter, the whiteness was maintained with the increasing storage time. However, the lowest value was found at day 12 in the VAC samples (*p* < 0.05). The non-enzymatic browning reaction between the lipid oxidation products and free amino groups of the surimi protein liberated by spoilage bacteria [56] could alter the color of the surimi gel during prolonged storage. Thus, the addition of DCE-60 partially retained the whiteness of the surimi gel packed under VAC conditions during storage at 4 °C. Nevertheless, the chlorophyll in the extract must be removed to avoid the negative effect on the whiteness or color of the resulting gel.

## 4. Conclusions

Duea ching fruit extract (DCE) prepared using 60% ethanol, namely DCE-60, had the highest total phenolic contents and antioxidant activity. DCE-60 at 0.05% levels enhanced the gelling properties by improving the breaking force, deformation and water holding capacity of the sardine surimi gel. Nevertheless, the whiteness of the gel sample (D60-0.05) was decreased. The addition of 0.05% DCE-60 lowered the deterioration of the gel properties for 12 days during refrigerated storage, especially when packed under vacuum condition as compared to modified atmosphere packaging. Moreover, 0.05% DCE-60 in combination with vacuum packaging lowered the microbial growth and lipid oxidation of the gel during storage at 4 °C. Therefore, DCE-60 at 0.05% could be used as a food-grade additive for surimi with no adverse impact on its acceptability.

## Figures and Tables

**Figure 1 foods-12-01635-f001:**
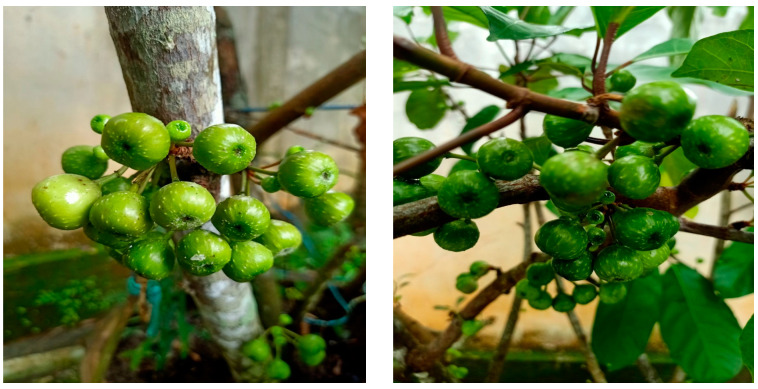
Duea ching plant bearing fruits.

**Figure 2 foods-12-01635-f002:**
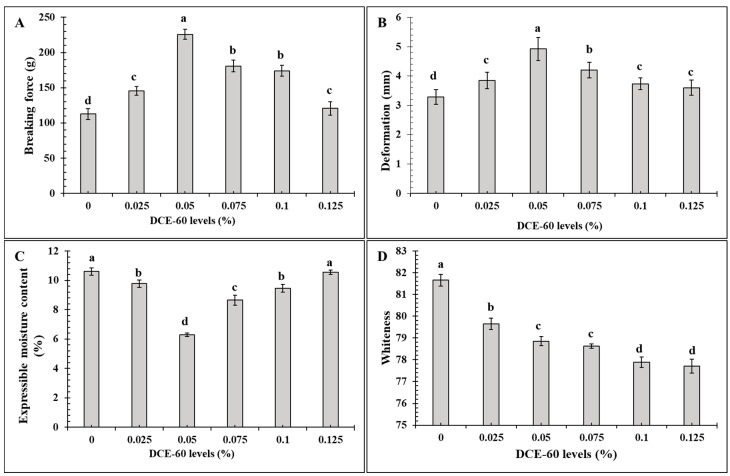
Breaking force (**A**), deformation (**B**), expressible moisture content (**C**) and whiteness (**D**) of the surimi gel added without and with DCE-60 at different levels. The bars represent the standard deviation (*n* = 3). The different lowercase letters on the bar indicate significant differences (*p* < 0.05). DCE-60: Duea ching extract prepared using 60% (*v/v*) ethanol.

**Figure 3 foods-12-01635-f003:**
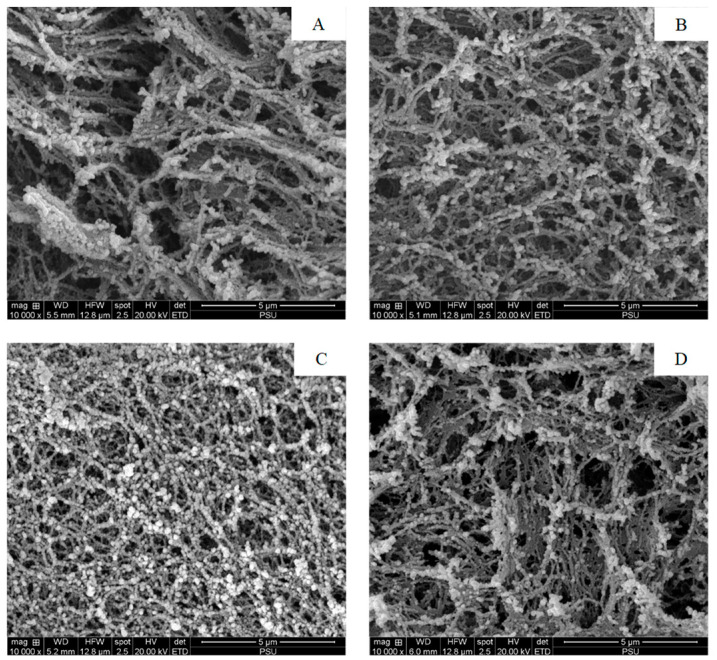
Electron microscopic images of the surimi gel without (**A**), and with 0.025% (**B**), 0.050% (**C**), and 0.125% (**D**) DCE-60 added. DCE-60: Duea ching extract prepared using 60% (*v/v*) ethanol. Magnification: 10,000×.

**Table 1 foods-12-01635-t001:** Total phenolic content, calcium content and antioxidative activities of the Duea ching extract prepared using distilled water and ethanol at various concentrations.

Sample	Ethanol Concentration (%, *v/v*)	Total Phenolic Content (mg GAE/g Dry Extract)	Calcium Content (mg/kg Dry Extract)	A-RSA (µmol TE/g Dry Extract)	D-RSA (µmolTE/g Dry Extract)	FRAP (µmol TE/g Dry Extract)	MCA (µmol EE/g Dry Extract)
DCE-W	0	308.51 ± 2.11 d	1082.03 ± 4.96 c	191.44 ± 4.17 e	93.32 ± 2.92 e	114.90 ± 2.18 e	17.94 ± 0.20 c
DCE-20 *	20	411.14 ± 2.96 c	1089.01 ± 4.04 b	220.65 ± 4.59 d	101.54 ± 3.09 d	140.61 ± 3.01 c	18.13 ± 0.10 c
DCE-40	40	617.83 ± 3.07 b	1090.23 ± 5.88 b	369.05 ± 5.01 c	175.74 ± 3.36 c	251.87 ± 2.05 b	20.32 ± 0.13 b
DCE-60	60	722.53 ± 2.82 a	1099.17 ± 6.21 a	501.48 ± 5.23 a	212.69 ± 4.00 a	312.47 ± 2.57 a	21.19 ± 0.09 a
DCE-80	80	615.58 ± 3.53 b	1092.12 ± 6.58 b	423.77 ± 4.93 b	189.36 ± 3.18 b	121.63 ± 2.79 d	20.35 ± 0.10 b
DCE-100	100	310.19 ± 1.96 d	1087.81 ± 5.59 b	212.01 ± 5.00 d	102.07 ± 2.74 d	116.00 ± 2.94 e	18.01 ± 0.17 c

Data are presented as mean ± SD (*n* = 3). Different lowercase letters in the same column indicates significant differences (*p* < 0.05). A-RSA and D-RSA: ABTS and DPPH radical-scavenging activities, respectively, FRAP: ferric reducing antioxidant power, MCA: metal chelating activity. DCE: Duea ching extract. DCE-W: Duea ching extract prepared using distilled water. * Numbers indicate ethanol concentrations (20–100%).

**Table 2 foods-12-01635-t002:** LC-MS data of the compounds present in the Duea ching extract prepared using 60% (*v/v*) ethanol (DCE-60).

Compound Name	Formula	*m*/*z*	Mass (g/mol)	Abundance (×10^6^)
Negative ion mode analysis				
Quinic acid	C_7_ H_12_ O_6_	191.06	192.06	1.43
Procyanidin B2	C_30_ H_26_ O_12_	577.13	578.14	0.55
Hydroquinone	C_6_ H_6_ O_2_	109.03	110.04	0.21
Indole-4-carboaldehyde	C_9_ H_7_ NO	144.05	145.05	2.04
Chlorogenic acid	C_16_ H_18_ O_9_	353.09	354.10	1.72
(±)-Catechin	C_15_ H_14_ O_6_	289.07	290.08	0.58
2,5-Dihydroxybenzaldehyde	C_7_ H_6_ O_3_	137.03	138.03	0.36
4-Acetoxyphenol	C_8_ H_8_ O_3_	151.04	152.05	1.05
Dihydroxyphenylacetic acid	C_8_ H_8_ O_4_	167.04	168.04	0.65
(±)-Taxifolin	C_15_ H_12_ O_7_	303.05	304.06	0.26
Rutin	C_27_ H_30_ O_16_	609.15	610.15	2.34
Isovitexin	C_21_ H_20_ O_10_	431.10	432.11	0.58
Quercetin 3-galactoside	C_21_ H_20_ O_12_	463.09	464.10	2.74
Umbelliferone	C_9_ H_6_ O_3_	161.02	162.03	0.16
Naringenin-7-O-Glucoside	C_21_ H_22_ O_10_	433.12	434.12	8.41
Methyl N-(α-methylbutyryl) glycine	C_9_ H_16_ O_4_	187.10	188.11	0.23
Abscisic acid (cis, trans)	C_15_ H_20_ O_4_	263.13	264.14	0.27
(±)-Naringenin	C_15_ H_12_ O_5_	271.06	272.07	1.41
Positive ion mode analysis				
Procyanidin B2	C_30_ H_26_ O_12_	579.15	578.14	0.08
Rutin	C_27_ H_30_ O_16_	611.16	610.15	0.23
Isovitexin	C_21_ H_20_ O_10_	433.11	432.11	0.23
Quercetin	C_15_ H_10_ O_7_	303.05	302.04	0.15
2,3-dinor-8-iso-PGF2a	C_18_ H_30_ O_5_	349.20	326.21	0.24

**Table 3 foods-12-01635-t003:** Textural profile of sardine surimi gel added without and with DCE-60 at different levels.

DCE-60 Levels (%)	Hardness (N)	Springiness (cm)	Cohesiveness (Ratio)	Gumminess (N)	Chewiness (N·cm)
0	171.79 ± 1.44 d	0.31 ± 0.04 d	0.36 ± 0.06 c	60.35 ± 0.90 d	60.93 ± 0.47 c
0.025	202.78 ± 2.01 b	0.34 ± 0.06 c	0.42 ± 0.04 b	112.11 ± 1.47 b	94.70 ± 0.59 a
0.050	250.09 ± 1.96 a	0.56 ± 0.02 a	0.50 ± 0.02 a	141.10 ± 1.81 a	95.46 ± 0.52 a
0.075	198.63 ± 1.46 c	0.43 ± 0.03 b	0.36 ± 0.03 c	72.72 ± 1.41 c	72.31 ± 0.67 b
0.100	173.65 ± 3.98 d	0.39 ± 0.05 c	0.33 ± 0.03 c	60.84 ± 1.54 d	71.59 ± 0.64 b
0.125	164.52 ± 5.19 e	0.32 ± 0.02 d	0.33 ± 0.02 c	55.13 ± 1.43 e	61.25 ± 0.54 c

The data are presented as mean ± SD (*n* = 3). The different lowercase letters in the same column indicate significant differences (*p* < 0.05). DCE-60: Duea ching extract prepared using 60% (*v/v*) ethanol.

**Table 4 foods-12-01635-t004:** Acceptability score for the surimi gel without and with DCE-60 added at different levels.

DCE-60 Levels (%)	Appearance	Color	Odor	Taste	Texture	Overall
0	6.88 ± 0.44 a	6.98 ± 0.34 a	6.36 ± 0.36 a	7.04 ± 0.29 a	6.25 ± 0.30 b	6.29 ± 0.27 bc
0.025	6.90 ± 0.41 a	6.82 ± 0.39 a	6.42 ± 0.34 a	7.01 ± 0.27 a	6.40 ± 0.40 ab	6.41 ± 0.22 b
0.050	6.89 ± 0.33 a	6.83 ± 0.44 a	6.50 ± 0.42 a	6.93 ± 0.31 a	6.95 ± 0.26 a	6.92 ± 0.34 a
0.075	6.84 ± 0.36 a	6.61 ± 0.32 a b	6.36 ± 0.43 a	6.72 ± 0.25 a	6.83 ± 0.22 a	6.77 ± 0.28 a
0.100	6.55 ± 0.38 ab	6.10 ± 0.35 b	6.33 ± 0.43 a	6.21 ± 0.29 b	6.24 ± 0.34 b	6.21 ± 0.30 bc
0.125	6.22 ± 0.30 b	6.01 ± 0.43 b	6.33 ± 0.42 a	6.07 ± 0.22 b	6.19 ± 0.30 b	6.08 ± 0.31 c

The data are presented as mean ± SD (*n* = 80). The different lowercase letters in the same column indicate significant differences (*p* < 0.05). DCE-60: Duea ching extract prepared using ethanol 60% (*v/v*).

**Table 5 foods-12-01635-t005:** Total viable count, psychrophilic bacteria count, peroxide value and TBARS of surimi gel without and with 0.05% DCE-60 added under different packaging conditions during refrigerated storage.

Packaging Condition	Storage Time (Days)	Total Viable Count (log CFU/g Surimi Gel)		Psychrophilic Bacteria Count (log CFU/g Surimi Gel)		Peroxide Value (mg Hydroperoxide Equivalents/ kg Surimi Gel)		TBARS (mg MDA/kg Surimi Gel)	
		Control	D60-0.05	Control	D60-0.05	Control	D60-0.05	Control	D60-0.05
	0	ND	ND	ND	ND	20.32 ± 0.09 Aex	20.27 ± 0.07 Aex	3.08 ± 0.03 Aex	3.05 ± 0.02 Afx
	2	2.34 ± 0.07 Adx	2.28 ± 0.11 Ae #	2.22 ± 0.03 Adx	2.18 ± 0.02 Aex	23.91 ± 0.05 Adx	21.11 ± 0.09 Bex	4.25 ± 0.03 Adx	3.14 ± 0.04 Bex
	4	2.89 ± 0.09 Acx	2.53 ± 0.08 Bdx	2.79 ± 0.06 Acx	2.44 ± 0.02 Bdx	40.69 ± 0.11 Abx	29.02 ± 0.04 Bdx	4.97 ± 0.02 Acx	3.72 ± 0.04 Bdx
ATM	6	4.36 ± 0.10 Abx	3.87 ± 0.09 Bcx	3.96 ± 0.04 Abx	3.71 ± 0.01 Bcx	45.38 ± 0.08 Aax	33.36 ± 0.09 Bcx	8.13 ± 0.03 Abx	4.01 ± 0.02 Bcx
	8	5.84 ± 0.07 Aax	4.99 ± 0.10 Bbx	5.62 ± 0.03 Aax	4.83 ± 0.03 Bbx	38.66 ± 0.08 Acx	40.12 ± 0.10B Ax	10.46 ± 0.04 Aax	7.56 ± 0.05 Bbx
	10	>6	5.68 ± 0.07 * ax	>6	5.63 ± 0.02 * ax	−	38.59 ± 0.09 * bx	−	8.03 ± 0.06 * ax
	12	>6	>6	>6	>6	−	−	−	−
VAC	0	ND	ND	ND	ND	20.30 ± 0.07 Afx	20.25 ± 0.06 Acx	3.05 ± 0.02 Adx	3.01 ± 0.04 Acx
	2	1.33 ± 0.08 * fz	ND	1.16 ± 0.04 *fz	ND	20.77 ± 0.03 Aez	20.35 ± 0.04 Bcy	3.09 ± 0.05 Adz	3.00 ± 0.04 Acy
	4	1.49 ± 0.07 Aez	1.40 ± 0.07 Adz	1.65 ± 0.09 Aez	1.37 ± 0.08 Bez	21.49 ± 0.08 Adz	20.99 ± 0.02 Bbz	3.21 ± 0.05 Acz	3.09 ± 0.05 Bbz
	6	1.83 ± 0.09 Adz	1.44 ± 0.08 Adz	2.15 ± 0.08 Adz	1.79 ± 0.10 Bdz	22.06 ± 0.12 Acz	21.07 ± 0.10 Bbz	3.25 ± 0.07 Acz	3.11 ± 0.03 Bbz
	8	2.60 ± 0.08 Acz	2.38 ± 0.10 Bcz	2.49 ± 0.10 Acz	2.24 ± 0.09 Bcz	22.39 ± 0.11 Acz	21.34 ± 0.08 Baz	3.26 ± 0.03 Acz	3.14 ± 0.03 Bbz
	10	3.74 ± 0.11 Aby	3.62 ± 0.12 Abz	3.61 ± 0.08 Abz	3.35 ± 0.07 Bbz	23.15 ± 0.09 Aby	21.41 ± 0.07 Baz	3.69 ± 0.06 Aby	3.22 ± 0.02 Baz
	12	5.17 ± 0.14 Aa ^#^	4.59 ± 0.09B Ay	4.98 ± 0.09 Aa ^#^	4.37 ± 0.11B Ay	25.67 ± 0.06 Aa ^#^	20.38 ± 0.07 Bcy	4.01 ± 0.04 Aa ^#^	3.23 ± 0.05 Bay
MAP	0	ND	ND	ND	ND	20.54 ± 0.10 Afx	20.46 ± 0.11 Af	3.05 ± 0.06 Aex	3.01 ± 0.04 Aex
	2	2.00 ± 0.13 * ey	ND	1.82 ± 0.12 *ey	ND	21.26 ± 0.15 Aey	20.51 ± 0.08 Bfy	3.12 ± 0.04 Aey	3.05 ± 0.03 Bdey
	4	2.35 ± 0.10 Ady	2.10 ± 0.08 Bey	2.29 ± 0.10 Ady	1.91 ± 0.11 Bey	24.45 ± 0.08 Ady	21.37 ± 0.09 Bey	3.36 ± 0.09 Ady	3.11 ± 0.08 Bdy
	6	2.60 ± 0.11 Acy	2.42 ± 0.14 Bdy	2.47 ± 0.09 Acy	2.30 ± 0.09 Bdy	32.22 ± 0.09 Acy	25.61 ± 0.12 Bdy	4.52 ± 0.12 Acy	3.14 ± 0.06 Bdy
	8	4.45 ± 0.13 Aby	3.86 ± 0.13 Bcy	3.98 ± 0.12 Aby	3.19 ± 0.13 Bcy	38.91 ± 0.14 Aay	29.36 ± 0.15Bcy	6.91 ± 0.13 Aby	3.48 ± 0.09 Bcy
	10	5.86 ± 0.09 Aax	4.79 ± 0.09 Bby	5.39 ± 0.14 Aay	4.05 ± 0.08 Bby	35.17 ± 0.11 Abx	34.96 ± 0.11 Aby	7.66 ± 0.08 Aax	4.05 ± 0.12 Bby
	12	−	5.80 ± 0.11 * ax	−	5.74 ± 0.11 * ax	−	38.04 ± 0.13 * ax	−	5.52 ± 0.08 * ax

The data are presented as mean ± SD (*n* = 3). The different uppercase letters (A and B) in the same row within the same parameter tested indicate significant differences (*p* < 0.05). The different lowercase letters (a, b…f) in the same column within the same packaging condition indicates significant differences (*p* < 0.05). The different lowercase letters (x, y and z) indicate significant differences among samples stored under different packaging conditions at the same storage time (*p* < 0.05). ATM: atmospheric packaging; VAC: vacuum packaging; MAP: modified atmospheric packaging; EMC: expressible moisture content; Control: surimi gel without DCE-60 powder added; D60-0.05: surimi gel with 0.05% DCE-60 powder added. * # sample not available for comparison. ND: not detected.

**Table 6 foods-12-01635-t006:** Breaking force, deformation, expressible moisture content (EMC) and whiteness of the surimi gel without and with 0.05% DCE-60 added under different packaging conditions during refrigerated storage.

Packaging Condition	Storage Time (Days)	Breaking Force (g)		Deformation (mm)		EMC (%)		Whiteness	
		Control	D60-0.05	Control	D60-0.05	Control	D60-0.05	Control	D60-0.05
ATM	0	108.91 ± 5.49 Bax	220.34 ± 6.10 Aax	3.25 ± 0.17 Bax	5.00 ± 0.16 Aax	10.63 ± 0.24 Adx	6.29 ± 0.18 Bfx	82.66 ± 0.27 Aax	78.85 ± 0.21 Bax
	2	107.04 ± 4.97 Bax	207.89 ± 5.77 Aby	3.19 ± 0.11 Bay	4.82 ± 0.15 Aay	10.79 ± 0.19 Adx	7.38 ± 0.20 Bex	81.97 ± 0.18 Aax	77.74 ± 0.19 Bax
	4	100.12 ± 5.38 Bax	175.36 ± 5.93 Acy	3.10 ± 0.10 Bay	4.26 ± 0.11 Aby	11.38 ± 0.22 Acx	8.74 ± 0.14 Bdx	81.10 ± 0.21 Aby	77.96 ± 0.20 Bbx
	6	87.88 ± 5.08 Bbxy	145.14 ± 5.23 Ady	2.84 ± 0.09 Bby	4.00 ± 0.08 Acz	18.59 ± 0.20 Abx	9.90 ± 0.17 Bcx	76.34 ± 0.22 Acz	76.66 ± 0.15 Acx
	8	70.14 ± 4.22 Bcz	100.66 ± 6.15 Aez	2.25 ± 0.14 Bcy	3.36 ± 0.17 Adz	22.44 ± 0.25 Aaz	12.87 ± 0.21 Bbx	75.17 ± 0.25 Adz	74.12 ± 0.18 Bdx
	10	−	78.81 ± 5.36 Afz	−	2.91 ± 0.14 * ez	−	15.94 ± 0.20 * ax	−	74.00 ± 0.20 * dx
	12	−	−	−	−	−	−	−	−
VAC	0	104.08 ± 7.33 Bax	213.04 ± 4.75 Aax	3.36 ± 0.12 Bax	5.02 ± 0.19 Aax	10.84 ± 0.20 Aex	6.18 ± 0.16 Bex	83.24 ± 0.17 Aax	77.89 ± 0.26 Bax
	2	106.78 ± 8.25 Bax	216.91 ± 4.83 Aax	3.31 ± 0.09 Bax	4.97 ± 0.11 Aax	10.89 ± 0.19 Aex	6.21 ± 0.17 Bey	82.99 ± 0.20 Aax	77.04 ± 0.16 Bax
	4	106.96 ± 6.24 Bax	201.78 ± 5.64 Aax	3.28 ± 0.15 Bax	4.91 ± 0.15 Aax	11.01 ± 0.21 Aey	6.38 ± 0.10 Bez	81.55 ± 0.24 Aax	76.83 ± 0.20 Bax
	6	92.24 ± 5.04 Bby	188.26 ± 6.69 Abx	3.07 ± 0.13 Babx	4.88 ± 0.14 Aax	12.13 ± 0.15 Ady	7.05 ± 0.14 Bdz	81.00 ± 0.15 Abx	76.75 ± 0.17 Bax
	8	91.35 ± 5.30 Bbx	173.34 ± 5.95 Acx	3.01 ± 0.11 Babx	4.84 ± 0.18 Aax	12.96 ± 0.19 Acz	7.46 ± 0.11 Bcz	80.32 ± 0.14 Acx	74.34 ± 0.19 Bbx
	10	83.08 ± 4.48 Bcx	170.81 ± 6.44 Acx	2.97 ± 0.14 Bbx	4.79 ± 0.13 Aabx	14.73 ± 0.18 Abz	7.91 ± 0.20 Bbz	77.21 ± 0.20 Adx	74.05 ± 0.21 Bbx
	12	72.19 ± 5.61 Bd ^#^	167.54 ± 5.16 Acx	2.55 ± 0.15 Bc #	4.63 ± 0.17 Abx	17.38 ± 0.22 Aa ^#^	9.43 ± 0.15 Bay	76.17 ± 0.19 Ae ^#^	73.91 ± 0.17 Bbx
MAP	0	106.38 ± 6.62 Bax	215.30 ± 5.26 Aax	3.24 ± 0.08 Bax	5.10 ± 0.14 Aax	10.75 ± 0.24 Adx	6.20 ± 0.09 Bfx	81.29 ± 0.18 Aax	77.90 ± 0.26 Bax
	2	104.37 ± 6.08 Bax	210.22 ± 6.01 Aax	3.21 ± 0.11 Bay	5.05 ± 0.09 Aax	10.77 ± 0.20 Adx	6.18 ± 0.10 Bfy	81.97 ± 0.12 Aay	77.75 ± 0.28 Bax
	4	103.91 ± 5.90 Bax	203.67 ± 6.22 Aax	3.21 ± 0.10 Bax	5.01 ± 0.11 Aax	10.84 ± 0.19 Ady	6.51 ± 0.11 Bey	80.44 ± 0.18 Aby	77.20 ± 0.14 Bax
	6	90.91 ± 4.17 Bby	190.34 ± 5.38 Abx	3.16 ± 0.06 Bax	4.64 ± 0.12 Aby	12.06 ± 0.13 Acy	6.99 ± 0.13 Bdy	77.14 ± 0.20 Acy	74.01 ± 0.11 Bby
	8	82.07 ± 5.36 Bcy	164.21 ± 5.74 Acy	3.00 ± 0.09 Bbx	4.35 ± 0.09 Acy	17.13 ± 0.14 Aby	7.59 ± 0.18 Bcy	76.05 ± 0.19 Ady	73.85 ± 0.19 Bcy
	10	75.15 ± 4.49 Bdy	150.19 ± 6.13 Ady	2.75 ± 0.08 Bcy	4.09 ± 0.11 Ady	20.69 ± 0.20 Aay	8.48 ± 0.10 Bby	73.30 ± 0.13 Aey	73.14 ± 0.21 Acy
	12	−	141.06 ± 5.69 * ey	−	3.22 ± 6.10 * ey	−	10.76 ± 0.12 * ax	−	73.02 ± 0.18 * cx

The data are presented as mean ± SD (*n* = 3). The different uppercase letters (A and B) in the same row within the same parameter tested indicate significant differences (*p* < 0.05). The different lowercase letters (a, b…f) in the same column within the same packaging condition indicates significant differences (*p* < 0.05). The different lowercase letters (x, y and z) indicate significant differences among the samples stored under different packaging conditions at the same storage time (*p* < 0.05). ATM: atmospheric packaging; VAC: vacuum packaging; MAP: modified atmospheric packaging; EMC: expressible moisture content; Control: surimi gel without DCE-60 powder added; D60-0.05: Surimi gel with 0.05% DCE-60 powder added. * # sample not available for comparison.

## Data Availability

The data are not shared.

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
