# Peer review of "Ethanolic Extract of Duea Ching Fruit: Extraction, Characterization and Its Effect on the Properties and Storage Stability of Sardine Surimi Gel"

_foods, 2023, doi:10.3390/foods12081635_

Round 1

Reviewer 1 Report

This is an interesting manuscript dealing with the effects of a natural extract on the characteristics of the surimi gel during storage which has a practical subject.

I think that the title of the manuscript is too long and can be shortened.

L33-34: correct the 4°C

L39: The scientific name should be in italic form.

In the first part of the introduction which is related to fig extract, I think that the authors should mention some studies related to the application of fig extract in food products and formulations.

L56: please correct the Ca2+ (the 2+ should be superscript)

L63-65: need to be rewritten.

L73: what are other additives for protein polymerization in surimi gels?

In the section 2.2.2.3, I think that the antioxidant activity test should be briefly explained.

In 2.3.2.1: please add the condition of test. Did you use the penetration test or TPA? Add the information such as the type of probe and speed.

L152: change the “effect of DCE extract” to “effect of DCE on”.

The discussion part is well written but the quality of this section can be improved by adding more studies in the field.

The ref #41 needs to be corrected.

Author Response

Reviewer 1

This is an interesting manuscript dealing with the effects of a natural extract on the characteristics of the surimi gel during storage which has a practical subject.

** Thank you for understanding in our works. The reviewer’s valuable comments are highly appreciated. All queries have been responded and the corrections have been made as track changes.

I think that the title of the manuscript is too long and can be shortened.

** The title of manuscript has been shortened but still informative as per the reviewer’s suggestion.

L33-34: correct the 4°C

** Corrected (Line 33).

L39: The scientific name should be in italic form.

** The scientific name has been changed into italic throughout the text. Thank you.

In the first part of the introduction which is related to fig extract, I think that the authors should mention some studies related to the application of fig extract in food products and formulations.

** To our knowledge, there is no report on the use of fig extract in food product directly, especially as the additives or ingredient for surimi gel strengthening. This is the first report on the use of fig extract, especially from the indigenous fig found in the tropical region. The fruits in the present study (Figure 1) were collected in the southern part of Thailand.

Authors plan to extend the utilization of this extract as additives such as antioxidant or antimicrobial agent in the near future.

L56: please correct the Ca2+ (the 2+ should be superscript)

** Corrected (Line 60). Also, the authors have been checked throughout the manuscript.

L63-65: need to be rewritten.

** The sentence has been rewritten for better clarity (Line 68-70). Thank you.

L73: what are other additives for protein polymerization in surimi gels?

** The other additives for protein polymerization or cross-linker had already been included in Line 64.

In the section 2.2.2.3, I think that the antioxidant activity test should be briefly explained.

** The details of each antioxidant activity assay have been given following the reviewer’s suggestion. Please see line 124-158. 

In 2.3.2.1: please add the condition of test. Did you use the penetration test or TPA? Add the information such as the type of probe and speed.

** For breaking force and deformation, the penetration test was used. Type of probe and testing speed have been provided for section 2.3.2. See line 180-182.

For TPA, the necessary details have been given as shown in section 2.3.2.3. See line 188-190.

L152: change the “effect of DCE extract” to “effect of DCE on”.

** DCE has been used to replace “DCE extract” throughout the manuscript. Thank you for reviewer’s suggestion.

The discussion part is well written but the quality of this section can be improved by adding more studies in the field.

** Some relevant papers have been added for in ‘Results and discussion’ section, especially on the use of plant extracts for surimi gel strengthening, which was the main objective of the present study. In addition, other study related with the storage stability of surimi gel incorporated with plant extract have been also included for discussion (Line 450-452)

The ref #41 needs to be corrected.

** Corrected.

Reviewer 2 Report

The same designations must be used in one manuscript - the terms: "fig" or "due ching".

When an abbreviation is mentioned for the first time in a main text of manuscript, it needs to be deciphered.

 Abstract – The abstract now presents the results in detail and mentions one conclusion

I would recommend adding information to this sections:

* relevance of the study

* research object

* key defined parameters

* the purpose of this study

 2. Materials and Methods – I'm not sure if such a detailed breakdown is necessary:

2.

2.1.

2.1.1.

2.1.1.1.

Line 43 – Flavonoids are also polyphenols

Line 67 – Reference

Line 69 – Reference

Line 87 – Geographical location of Songkhla

Line 92 – What equipment was used to perform the extraction.

Line 93 – What Lasting of centrifugation

Line 95 – How and with what equipment?

Line 96 – Where were the samples collected, where were they packed?

Line 100 – How many replicates were performed for each extract?

Line 124 – What was the temperature and duration of the process.

Line 143 - Line 151 – There is something wrong with the numbering.

Line 150 – It is not quite clear how it was done.

Line 160 –How many repetitions of each type were packed? How many replicates were each type analyzed?

Line 174 –How many replicates were there for each extract? How many repetitions of each type were packed? How many replicates were each type analyzed? How many packages were used for each analysis? How many replicates were there for each analysis?

 3. Results and discussion – It is not necessary to duplicate numerical values both in the text of the manuscript and in the tables. I would recommend checking that all results that identify significant differences between samples are properly lowercase superscript.

Line 292 –Are the axes labeled correctly in Figure 2?

Line 330 – There are contradictions with the results of Table 3.

Line 332 – There are contradictions with the results of Table 3.

Line 334 – There are contradictions with the results of Table 3.

Line 345 – Or Table 4.

Starting from page 12, the page numbering is incorrect.

The reference to Table 6 cannot be found.

Line 464 – Or Table 6.

Line 501 – I would recommend using transcripts of abbreviations in the conclusions as well.

Author Response

Reviewer 2

The same designations must be used in one manuscript - the terms: "fig" or "due ching".

**Thank you for the comment. Term ‘due ching’ is the common name used to call the fruit used in the present study. Due ching is a type of indigenous fig found in the tropical region. The details have been given in line 38-40.

For the rest of text, the authors used ‘ethanolic due ching extrcat’ or ‘EDCE’ for consistency and better understanding.

When an abbreviation is mentioned for the first time in a main text of manuscript, it needs to be deciphered.

**Thank you so much. Authors have cross-checked all the abbreviations. The description has been given for all the abbreviations for the first appearance in the text.  

Abstract – The abstract now presents the results in detail and mentions one conclusion

I would recommend adding information to this sections:

* relevance of the study

* research object

* key defined parameters

* the purpose of this study

** The abstract has been improved for better clarity by including the information as guided by the reviewer starting from background and objective/purpose of the study. Please see line 21-25.

For the key parameters such as the composition and antioxidant activity of the extract as well as  gel property, those information had been already presented in the abstract. Thank you so much for the suggestion to make the abstract more informative and better understandable. 

  1. Materials and Methods – I'm not sure if such a detailed breakdown is necessary:

2.

2.1.

2.1.1.

2.1.1.1.

**Authors would like to provide the better flow by giving the sub-section under the major heading. Therefore, authors would like to keep this format for the ease to follow.

Line 43 – Flavonoids are also polyphenols

** The sentence has been rewritten (Line 48).

Line 67 – Reference

** Reference has been added (Line 72).

Line 69 – Reference

** Reference has been added (Line 74).

Line 87 – Geographical location of Songkhla

** Songkhla is located in southern part of Thailand. The detail has been added (Line 92-93).

Line 92 – What equipment was used to perform the extraction.

** A magnetic stirrer was used for the extraction. Brand and manufacturing country have been provided (Line 98).

Line 93 – What Lasting of centrifugation

** Time for centrifugation was 30 min. The detail has been included (Line 99).

Line 95 – How and with what equipment?

** Nitrogen purging was done by connecting one side of PVC air hose to nitrogen gas tank and another side was place in the headspace of container filled with the filtrate. The gas valve was opened to allow nitrogen to remove the residual ethanol in the filtrate. 

Line 96 – Where were the samples collected, where were they packed?

** The obtained extracts were placed in the amble bottle, capped tightly and stored in desiccator under dark condition until use or analysis. Please see line 104-105.

Line 100 – How many replicates were performed for each extract?

**All experiment and analysis were carried out in triplicate. The detials have been added in section ‘2.5. Statistical analysis’. Please see line 220-221.

Line 124 – What was the temperature and duration of the process.

** The temperature of surimi paste preparation was below 10 °C. See line 173-174. The duration from chopping to stuffing was less than 5 min. The time for chopping and mixing has already present in the text.

Line 143 - Line 151 – There is something wrong with the numbering.

** The numbering has been checked and corrected. Thank you.

Line 150 – It is not quite clear how it was done.

** Details of microstructure analysis have been given for better understanding (Line 198-201).

Line 160 –How many repetitions of each type were packed? How many replicates were each type analyzed?

** As mention in section ‘2.5. Statistical analysis’ All experiment and analysis were carried out in triplicate.

Line 174 –How many replicates were there for each extract? How many repetitions of each type were packed? How many replicates were each type analyzed? How many packages were used for each analysis? How many replicates were there for each analysis?

** As details in section ‘2.5. Statistical analysis’.

  1. Results and discussion – It is not necessary to duplicate numerical values both in the text of the manuscript and in the tables. I would recommend checking that all results that identify significant differences between samples are properly lowercase superscript.

** Thank you for the insightful comment. Authors understand the point raised by the reviewers. We only pointed out only a few major values in the text, not all the values since those values can be seen directly from the tables.

For the significant differences between the samples, we have changed from the lowercase superscript to ‘lowercase letters’ in the table footnoted and the figures.

Line 292 –Are the axes labeled correctly in Figure 2?

** Thank you for your suggestion. The axes labeled of Figure 2B and 2C have been corrected and the revised figure has been replaced.

Line 330 – There are contradictions with the results of Table 3.

Line 332 – There are contradictions with the results of Table 3.

Line 334 – There are contradictions with the results of Table 3.

** Thank you so much. Sorry for the mistake. We have cross-checked and corrected for the significant difference between different samples.

Line 345 – Or Table 4.

** The number of Table has been corrected. Sorry for the mistake.

Starting from page 12, the page numbering is incorrect.

** Thank you for an advice. Page number has been cross-checked and corrected.

The reference to Table 6 cannot be found.

** The reference to Table 6 has been addressed in Line 505.

Line 464 – Or Table 6.

** The sentence was referred to the microbial count and lipid oxidation. Thus, it should be written as Table 5 (Line 520).

Line 501 – I would recommend using transcripts of abbreviations in the conclusions as well.

** The conclusion has been edited according to the reviewer’s recommendation. Thank you.